Wavelet transform-based mode decomposition for EEG signals under general anesthesia

Yamochi Shoko 1
Yamada Tomomi 1
Obata Yurie 2
Sudo Kazuki 1
Kinoshita Mao 1
Akiyama Koichi 3
Sawa Teiji anesth@koto.kpu-m.ac.jp 4
1 Department of Anesthesiology, Kyoto Prefectural University of Medicine , Kyoto , Japan
2 Department of Anesthesia, Yodogawa Christian Hospital , Osaka , Japan
3 Department of Anesthesiology, Kindai University , Osakasayama , Osaka , Japan
4 University Hospital, Kyoto Prefectural University of Medicine , Kyoto , Japan
Harrison Neil
Electronic publication date: 2024 Nov 15
Publication date: 2024
Volume: 12
Electronic Location ID: e18518
Received 2024 Jul 11; Accepted 2024 Oct 22
Copyright: ©2024 Yamochi et al.
Copyright year: 2024
Copyright holder: Yamochi et al.
License: This is an open access article distributed under the terms of the Creative Commons Attribution License, which permits unrestricted use, distribution, reproduction and adaptation in any medium and for any purpose provided that it is properly attributed. For attribution, the original author(s), title, publication source (PeerJ) and either DOI or URL of the article must be cited.
License URL: https://creativecommons.org/licenses/by/4.0/

Keywords: Electroencephalogram, Variational mode decomposition, Empirical wavelet transform, Wavelet mode decomposition, Intrinsic mode function

Funding: The authors received no funding for this work.

==============================
Background

Mode decomposition methods are used to extract the characteristic intrinsic mode function (IMF) from various multidimensional time series signals. We analyzed an electroencephalogram (EEG) dataset for sevoflurane anesthesia using two wavelet transform-based mode decomposition methods, comprising the empirical wavelet transform (EWT) and wavelet mode decomposition (WMD) methods, and compared the results with those from the previously reported variational mode decomposition (VMD) method.

Methods

To acquire the EEG data, we used the software application EEG Analyzer, which enabled the recording of raw EEG signals via the serial interface of a bispectral index (BIS) monitor. We also created EEG mode decomposition software to perform empirical mode decomposition (EMD), VMD, EWT, and WMD operations.

Results

When decomposed into six IMFs, the EWT enables narrow band separation of the low-frequency bands IMF-1 to IMF-3, in which all central frequencies are less than 10 Hz. However, in the upper IMF of the high-frequency band, which has a center frequency of ≥ 10 Hz, the dispersion within the frequency band covered was widespread among the individual patients. In WMD, a narrow band of clinical interest is specified using a bandpass filter in a Meyer wavelet filter bank within a specific mode-decomposition discipline. When compared with the VMD and EWT methods, the IMF that was decomposed via WMD was accommodated in a narrow band with only a small variance for each patient. Multiple linear regression analyses demonstrated that the frequency characteristics of the IMFs obtained from WMD best tracked the changes in the BIS upon emergence from general anesthesia.

Conclusions

The WMD can be used to extract subtle frequency characteristics of EEGs that have been affected by general anesthesia, thus potentially providing better parameters for use in assessing the depth of general anesthesia.

Introduction

General anesthetics affect a patient’s central nervous system by reducing consciousness, as demonstrated by their altered electroencephalogram (EEG) patterns. These EEG patterns suggest that intraoperative EEG monitoring is vital for assessment of the effects of anesthetics on the brain (Ching et al., 2010; Flores et al., 2017; Purdon et al., 2015; Supp et al., 2011). EEG monitoring can help to minimize anesthetic use, reduce recovery times, and prevent accidental awakening during surgery. Advances in the clinical applications of EEG monitoring during general anesthesia (GA) are now focusing on technologies that decode the EEG data to gauge both anesthesia depth and consciousness levels, thus enhancing our understanding of brain function during GA and ensuring that appropriate anesthesia levels are maintained. Although several current anesthesia depth algorithms are not yet publicly available because they are proprietary, these algorithms are based on the principles of EEG frequency analysis using the Fourier transform.

In a different approach, the Hilbert–Huang transform (HHT), which was introduced in the 1990s (Huang et al., 1998), combines empirical mode decomposition (EMD) with the Hilbert transform and has been used in a variety of studies (Kortelainen & Vayrynen, 2015; Li et al., 2008; Liu et al., 2018; Shalbaf et al., 2012), including studies of propofol anesthesia (Obata et al., 2023). However, EMD lacks a solid mathematical foundation and this has led to the development of variational mode decomposition (VMD) as an alternative method (Dragomiretskiy & Zosso, 2014).

This study aims to capture the frequency characteristics of EEGs that fluctuate in tandem with changes in the level of patient consciousness—or the depth of anesthesia—in response to the dosage or concentration of the anesthetics used during GA. The goal of the study is to use these characteristics to evaluate the depth of anesthesia. Clinical devices such as the bispectral index (BIS value; BIS Monitor, Medtronic, Dublin, Ireland) and the patient state index (PSi) value (Masimo SedLine, Irvine, CA, USA), which rely on spectral analyses performed using Fourier transforms, are commonly used to assess the depth of anesthesia. However, the manufacturers have not disclosed the proprietary algorithms that lie behind these devices, including the calculation methods used. This lack of transparency is problematic, because scientific papers are being published based on parameters for which the underlying science remains unclear. The development of precise feature extraction methods that can calculate alternative parameters has the potential to improve the accuracy of GA management significantly. In this study, three different mode decomposition methods—VMD, the empirical wavelet transform (EWT), and wavelet mode decomposition (WMD)—were applied to an EEG dataset (Yamada et al., 2023) recorded over 30 min that captured the transition from anesthesia to wakefulness during sevoflurane GA. By analyzing the changes observed in the frequency characteristics of the decomposed intrinsic mode functions (IMFs), we aim to identify parameters that can contribute to more accurate evaluation of anesthesia depth.

Materials & Methods

Algorithm for the EWT and the Hilbert transform

In 2013, Gilles (2013) presented the EWT method, in which adaptive wavelet filter banks were built to decompose a given signal into IMFs. Like EMD, the EWT is intended to extract the IMF components from a signal. However, the EWT operates in the frequency domain, whereas EMD is performed in the time domain. Therefore, in the EWT, it is necessary to consider where the construction algorithm used for the filter banks draws boundaries in the frequency content within the spectrum of the signal under analysis (see Fig. 1 and Fig. S1). If we assume that the frequency domain [0, π] is divided into N consecutive segments, we must then extract N −1 boundaries, excluding the 0 and π boundaries. The local maxima in the spectrum are then determined and sorted in descending order to locate the boundaries for the EWT. Each boundary was defined as the average of the consecutive maxima. Let ωn be the limit between each segments (where ω0 =0 and ωN = π) and let each segment be denoted by Λn = [ ωn −1, ωn]; then, ⋃n=1NΛn= [0, π]. A transition phase Tn (τn = γωn, where 0 <γ <1) of width 2 τn is defined near the center of each Λn, as illustrated in (Figs S2A and S2B). An empirical wavelet is defined as a bandpass filter for each Λn (Figs S2C and S2S) (Daubechies, 1992). For this purpose, Gilles (2013) used the following steps when constructing the Meyer wavelets.

Figure 1 Algorithm flowchart for the EWT and WMD, including a figure that explains the construction of the Meyer wavelet filter bank.

LPF, low-pass filter; BFP, bandpass filter.

EWT steps

Step (1)

Apply a fast Fourier transform to the signal f (t)(t = {ti}, i=1,2,....,M, where M denotes the number of samples) to obtain the frequency spectrum X(ω). Then, find the set of maxima M ={Mi}i=1,2,...,N in the Fourier spectrum by applying the convolution integral and estimate their corresponding frequencies ω = {ωi}i=1,2....,N, where N denotes both the number of maxima and the number of filter banks introduced from this point.

Step (2)

Perform appropriate segmentation of the Fourier spectrum acquired in Step 1 and define the boundaries Ωi for each segment as the center points between two consecutive maxima: (1) Ωi=ωi+ωi+12

where ωi and ωi+1 are both frequencies and the set of boundaries is Ω = {Ωi}i=1,2,N−1.

Step (3)

The empirical scaling function and the empirical wavelets with ∀n >0 are defined in Eqs. (2) and (3), respectively, as shown in Fig. S2C, where ∀ denotes universal quantification and is read as “for all”. (2) ϕ ˆnω=ω≤ωn−τn1ωn−τn≤ω≤ωn+τn cosπ2β12τnω−ωn+τnotherwise0

(3) ψ ˆnω=ωn+τn≤ω≤ωn−τn1ωn+1−τn+1≤ω≤ωn+1+τn+1 cosπ2β12τn+1ω−ωn+1+τn+1ωn−τn≤ω≤ωn+τn sinπ2β12τnω−ωn+τnotherwise0

where β(x) is any function Ck ([0,1]), such that: (4) βx=x≤00x≥11x∈0,1x+β1−x=1

where (5) βx=x435−84x+70x2+20x3

Regarding the selection of τn, τn is proportional to ωn:τn = γωn, where 0 <γ < 1. Consequently, when ∀n >0, Eqs. (6) and (7) can be redefined as follows: (6) ϕ ˆnω=ω≤1−γωn11−γωn≤ω≤1+γωn cosπ2β12γωnω−1−γωnotherwise0

(7) ψ ˆnω=1+γωn≤ω≤1−γωn+111−γωn+1≤ω≤1+γωn+1 cosπ2β12γωn+1ω−1−γωn+1

Proposition 1: if γ<minnωn+1−ωnωn+1+ωn.

The concept for construction of the Meyer wavelet is based on the set ϕ1t,ψntn=1N, which is a tight frame when (8) ∑k=−∞+∞ϕ ˆ1ω+2kπ2+ ∑n=1Nψ ˆnω+2kπ2=1

as shown in Fig. S2D.

Step (4)

Empirical wavelet transform

The EWT Wfɛ∈n,t of the signal f (t) is defined as the inner product of the signal to be analyzed f with the scaling function and the empirical wavelets. (9) Wfɛ0,t=f,ϕ1= ∫fτϕ1τ−t¯dτ

(10) Wfɛn,t=f,ψn= ∫fτψnτ−t¯dτ

where Wfɛ(i, t) denotes the detailed coefficient of the i- th filter bank at a time t.

The total mode f (t) can then be reconstructed as follows: (12) ft=Wfℰ0,t∗,ϕ1t+∑n=1NWfℰn,t∗ψnt=Wfℰ ^0,ωϕ ˆ1ω+ ∑n=1NWfℰ ^n,ω∗ψ ˆnω∨

Here, ∧ denotes a Fourier transform and ∨ denotes an inverse Fourier transform.

Finally, the empirical mode fk is defined as follows: (13) f0t=Wfℰ0,t∗,ϕ1t

(14) fkt=Wfℰk,t∗,ψkt

Algorithm for WMD

In the EWT, the boundary frequencies that separate the IMFs are determined as variable values for each analysis epoch by searching for local polar regions of power via application of an empirical method in the Fourier domain. (15) EEG=IMF10.125Hz∼Ω1+IMF2Ω1∼Ω2+IMF3Ω2∼Ω3+IMF4Ω3∼Ω4+IMF5Ω4∼Ω5+IMF6Ω5∼64Hz

(where Ωi denotes the boundary between the (i −1)th segment and the i-th segment)

However, if the boundary frequencies are determined in advance and the transformation performed by the Meyer wavelet is used as a bandpass filter, the EEG can be then decomposed into multiple IMF modes that fall within the range of the target frequencies. This method is a modified EWT method; however, we call it the WMD method because it discards the empirical approach and decomposes the process into modes with specific target frequencies via the Meyer wavelet transform (see Fig. 1), as indicated by the following equation: (16) EEG=IMF10.125∼4Hz+IMF24∼8Hz+IMF38∼14Hz+IMF414∼20Hz+IMF520∼30Hz+IMF630∼64Hz.

To set the preset boundaries Ω1−5 required in Step 2 of the EWT, the EWT algorithm was modified to decompose six IMFs in the WMD. Ω1, Ω2, Ω3, Ω4, and Ω5 were set at 4 Hz, 8 Hz, 14 Hz, 20 Hz, and 30 Hz, respectively.

Anesthesia management and data acquisition

As reported previously (Yamada et al., 2023), the original EEG data were acquired according to the principles of the Declaration of Helsinki. The Institutional Review Board at the Kyoto Prefectural University of Medicine (IRB of KPUM) approved the current study (No. ERB-C-1074) for human experiments. The requirement for informed patient consent was waived by the IRB of the KPUM for this noninterventional and noninvasive retrospective observational study (all patients were provided with an opt-out option, of which they were notified in the preoperative anesthesia clinic). We analyzed the intraoperative EEG data (the sampling frequency is 128 Hz and the data length is 8 s, therefore, the number of data points per epoch is 1,024) acquired from 10 patients that were maintained under sevoflurane anesthesia. Data acquisition was performed via using EEG Analyzer software (ver. 54_GP; https://anesth-kpum.org/blog_ts/?p=3169) (Hayase et al., 2021) linked to a VISTA A-3000 BIS monitor with a BIS Quatro sensor. For further details about patient information, anesthesia management, and the data acquisition procedure, including raw EEG data, the BIS index, the 95% spectral edge frequency (SEF95), the total power (TP), and the absolute power derived from electromyography in the 70–110 Hz range (EMG low), please see our previous report (Yamada et al., 2023). This analysis used the sevoflurane GA dataset (Supplemental File S3) that we reported previously (Yamada et al., 2023).

EEG Mode Decomposition for EMD, VMD, EWT, and WMD

The EMD analysis software EEG Mode Decompositor (ver. 2.7N, https://anesth-kpum.org/blog_ts/?p=4114) was initially developed to perform EMD (Obata et al., 2023) and was subsequently updated by adding a VMD mode (Yamada et al., 2023) and, in this case, EWT and WMD modes (Gilles, 2013). Supplemental File S4 (Python programming code for EWT) shows the Python code that was described in the report by Carvalho et al. (2020) and used to perform EWT analysis. Supplemental File S5 (Processing programming code for EWT and WMD) shows the EWT and WMD classes of the Processing and Java programming codes, respectively. We intended to observe significant changes in specific narrow frequency bands within the 0–64 Hz range, such as δ (0.0625<f ≤4 Hz), θ (4<f ≤8 Hz), α (8<f ≤14 Hz), low- β (14<f ≤20 Hz), high- β (20<f ≤30 Hz), and γ waves (30<f ≤64 Hz).

Data processing and statistics

The changes in the various EEG parameters between two time points were compared statistically by performing Kruskal–Wallis tests with SPSS Statistics (ver. 28.0.1.0, IBM Corp., Armonk, NY, USA), and p values <0.05 were considered statistically significant. Multiple linear regression (MLR) analysis involving between the BIS index and the parameters of the IMFs was performed via the scikit-learn and StatsModel libraries in Python (ver 3.8), as we reported recently (Sawa, Yamada & Obata, 2022).

Results

Case analysis of the EWT application

First, we present the VMD, EWT, and WMD analyses that were conducted using the EEG data acquired from Patient #6 (a 27-year-old female with an ovarian tumor undergoing laparoscopic oophorectomy) (Yamada et al., 2023) when anesthetized with sevoflurane (video files for a 30 min period until emergence: Supplemental Files S6–S9: Sev6_EMD.mp4, Sev6_VMD.mp4, Sev6_EWT.mp4, and Sev6_WMD.mp4, respectively). For a comparative analysis with the previously reported VMD adaptation (see Figs. 3, and 4 in Yamada et al. 2023), we analyzed 4-s-long epochs of EEGs that were recorded at 128 Hz during three distinct phases of the GA up to the point of emergence: the maintenance (30 min before emergence), transition (1 min before emergence), and emergence phases. Figure 2 presents the raw EEGs (4 s), the IMFs obtained over 4 s via the three decomposition methods, and the density spectral arrays (DSAs) (64 s, starting with a 4-s-long segment of raw EEG data) across these GA phases. Given the challenges involved in visually discerning the differences among the characteristics of the IMFs from such waveform diagrams, a Hilbert transform was applied to derive both the instantaneous frequency (IF) and instantaneous amplitude (IA). The results are shown in the Hilbert spectrum in Fig. 3. During the maintenance phase of anesthesia, IMFs 1 to 4 were analyzed using the VMD method and were predominantly in the frequency region around 20 Hz or lower. In contrast, the six IMFs spanned a broader frequency range from 0 Hz up to 50–60 Hz in both the EWT and WMD methods. During the transition period, IMF-6 notably shifted noticeably to the 30-Hz frequency during the VMD analysis, whereas the peaks of IMF-5 and IMF-6 in the EWT analysis moved toward higher frequencies above 40 Hz. Upon emergence, the peaks of IMF-5 and IMF-6 in the EWT increased to frequencies above 50 Hz, whereas they remained below 40 Hz in the VMD results. The WMD method displays frequency histograms consistently for each IMF within a narrow band corresponding to predefined target frequencies regardless of the GA phase considered. This distinction arises because the EWT and VMD do not impose frequency band constraints on the decomposition of IMFs; in contrast, WMD requires decomposition into IMFs within specified frequency bands.

Figure 2 DSA, raw EEG, and IMFs after mode decomposition in Patient #6 (27-year-old woman) (Yamada et al., 2023).

Three anesthesia phases (maintenance, transition, and emergence) were analyzed using three mode decomposition methods. (A) DSA (spectrogram) of 64 s using a multitaper method. (B) 4-second raw EEG. (C) IMFs. (1) VMD, (2) EWT, and (3) WMD methods.

Figure 3 Hilbert spectra of six IMFs from mode decomposition of the 4-second EEG in Patient #6 (27-year-old woman) (Yamada et al., 2023).

Three anesthesia phases (maintenance, transition, and emergence) were analyzed using three mode decomposition methods: (A) VMD, (B) EWT, and (C) WMD methods.

Hilbert spectrograms of the IMFs in the EWT and WMD methods

To gain further understanding of the frequency characteristics of each IMF produced by VMD, the EWT, and WMD across the 30 min phase before emergence, a Hilbert spectral analysis of the IMFs was performed. This analysis assessed the variations in the IF and IA values derived from the Hilbert transforms of the IMFs during each phase for all ten patients. Fig. S3 shows the Hilbert spectrograms for each IMF obtained via the VMD, EWT, and WMD methods, the summed IMF (IMF-all, which corresponds to the original EEG), and the DSAs for 30 min before emergence for all patients. In the EWT method, IMF-1, IMF-2, and IMF-3 effectively capture the δ, θ, and α frequency bands, respectively. After emergence, however, a notable reduction in the power of IMF-3, which primarily extracts the α-band, was observed. The central frequencies of IMF-4, IMF-5, and IMF-6 span a broad spectrum within the β- and γ-bands. Conversely, in the WMD approach, IMF-1, IMF-2, and IMF-3 effectively extract the δ, θ, and α bands, respectively, and this is consistent with their preset boundaries. In contrast to the EWT results, IMF-4, IMF-5, and IMF-6 are characterized by narrow bandwidths within specific frequency ranges, i.e., low β, high β, and γ, respectively. Notably, the WMD method demonstrated more stable frequency band retention across the separated IMFs than either of the EWT and VMD methods.

Multiple linear regression models to predict the BIS

Next, we analyzed the processed EEG data, which were sampled every 8 s, and were acquired from 10 patients who had been administered a sevoflurane GA 30 min before emergence. We explored the relationship between the BIS index (a clinical gauge of the depth of anesthesia) and the IMF parameters, including the center frequency (i.e., the average of the Hilbert spectrum) and TP, which were converted from the power in P (µV2) to decibels (dB), using dB = 10 ⋅ log10P). MLR models were developed using the IMF parameters to predict the BIS index based on the Data_EME30s dataset (Supplemental File S11). These models incorporated the center frequencies and TP values of the IMFs as variables and were tested using three modal decomposition methods: VMD, EWT, and WMD. The comprehensive MLR formula included individual terms for the center frequencies and the TPs of the six IMFs, thus leading to a model with 36 explanatory variables. Figure 4 shows the time-trend graphs for the BIS index, the SEF95, the TP, EMGlow, and the corresponding IMFs and TPs across the decomposition methods. The analysis yielded robust correlations, with the WMD-based MLR model using all 36 parameters (18 center frequencies and 18 TPs) and demonstrating the strongest correlation (R 2 = 0.899) and the lowest errors (mean absolute error: MAE = 0.244; root mean squared error: RMSE = 0.318) among the four models examined; the results are detailed in Table 1. A simplified WMD model that used only 12 parameters ranked second. For a more targeted approach, we evaluated an MLR model using only seven IMF parameters (IMF-1_freq, IMF-3_freq, IMF-5_freq, IMF-6_freq, IMF-2_TP, IMF-4_TP, and IMF-6_TP) that presented significant p values (less than 0.05) but still presented excellent fit metrics (R 2 =0.897, MAE = 0.244, RMSE = 0.321), as shown in Fig. 5 and Table S1. Further analyses were conducted by using separate models for the central frequencies and TPs, and the results confirmed that combining the two parameters offered superior predictive accuracy, as shown in Fig. S4 and Table S2. Notably, in the WMD approach, the center frequencies of IMF-1, IMF-3, IMF-5, and IMF-6, and the TPs of IMF-2, IMF-4, and IMF-6, were identified as significant predictors. This finding indicates that WMD effectively captures the narrow band changes in the IMF parameters correlated with the shifts in the BIS index that occur during emergence from GA, thus indicating the superior sensitivity of WMD for monitoring changes in depth of anesthesia.

Figure 4 Time course changes of EEG parameters.

(A) Processed EEG parameters (BIS index, SEF95, total power, and EMGlow), (B) Central frequencies, and (C) Total powers of the intrinsic mode functions (IMFs) of the variational mode decomposition (VMD), empirical wavelet transform (EWT), and wavelet mode decomposition (WMD) methods for 30 min before emergence from GA in ten patients (Yamada et al., 2023). The total power is displayed on a log scale (in decibels (dB); obtained from the power (P, µV 2) by using the conversion dB = 10 × log10P). The data are presented as medians (dark blue lines), 25th–75th percentile ranges (light blue areas), and max–min ranges (grey areas). ∗p < 0.05 between the value at –30 min and the value at the emergence, based on the Kruskal–Wallis test. Processed EEG parameters and VMD data were reconstructed as panel diagrams based on data from our previous report (Figs. 5, 6 of Yamada et al. 2023). BIS index: bispectral index, SEF95: spectral edge frequency 95%, EMGlow: absolute power derived from electromyography in the 70–110 Hz range, EM, emergence.

Table 1 MLR analysis of the BIS values and the parameters of the IMFs in VMD, the EWT, or WMD.

	(A) VMD	(B) EWT	(C) WMD	
determination factor	0.837	0.618	0.898	
MAE	0.313	0.459	0.244	
RMSE	0.403	0.618	0.319	
y-intercept	9.707e−17	−6.533e−16	1.170e−16	
explanatory variables	regression coefficient (P>—t—)	
central frequency  IMF-1	−0.0400 (0.239)	0.0863 (0.001*)	0.0863 (0.041*)	
IMF-2	0.0334 (0.454)	0.0223 (0.973)	0.0223 (0.330)	
IMF-3	−0.0203 (0.731)	−0.0674 (0.001*)	−0.0674 (0.022*)	
IMF-4	0.0812 (0.370)	0.0180 (0.001*)	0.0180 (0.647)	
IMF-5	0.0332 (0.758)	0.203 (0.889)	0.2035 (0.000*)	
IMF-6	0.3371 (0.000*)	0.141 (0.905)	0.145 (0.019*)	
total power  IMF-1	0.1201 (0.004*)	0.0211 (0.202)	0.0211 (0.472)	
IMF-2	−0.0433 (0.323)	−0.270 (0.001*)	−0.2797 (0.000*)	
IMF-3	−0.1049 (0.066)	0.0246 (0.252)	0.0246 (0.719)	
IMF-4	−0.2488 (0.000*)	−0.255 (0.008*)	−0.2554 (0.000*)	
IMF-5	−0.0086 (0.860)	0.0303 (0.001*)	0.0303 (0.474)	
IMF-6	−0.0753 (0.087)	0.281 (0.292)	0.2812 (0.000*)	
	(D) VMD+EWT+WMD	
determination factor	0.899	
MAE	0.244	
RMSE	0.318	
y-intercept	−2.75e−16	
explanatory variables	EWT, reg. coef. (P>—t—)	WMD, reg. coef. (P>—t—)	VMD, reg. coef. (P>—t—)	
central frequency  IMF-1	0.1763(0.010*)	−0.0125 (0.868)	−0.0298 (0.452)	
IMF-2	0.0683 (0.109)	−0.0210 (0.481)	0.1302 (0.023*)	
IMF-3	−0.1435 (0.054)	−0.0782 (0.027*)	−0.0127 (0.861)	
IMF-4	0.1666(0.001*)	0.0595 (0.158)	0.0028 (0.975)	
IMF-5	0.0217 (0.559)	0.1124 (0.056)	0.0076 (0.933)	
IMF-6	−0.0077 (0.809)	0.1595 (0.020*)	0.1013 (0.251)	
total power IMF-1	0.0557(0.677)	−0.1463 (0.245)	0.0926 (0.091)	
IMF-2	0.0153 (0.732)	−0.2335 (0.002*)	0.1147 (0.085)	
IMF-3	0.0196 (0.610)	−0.0549 (0.519)	−0.0563 (0.322)	
IMF-4	0.1085 (0.011*)	−0.2700(0.000*)	−0.0305 (0.640)	
IMF-5	0.0868 (0.029*)	0.0254 (0.613)	0.0005 (0.993)	
IMF-6	0.0093 (0.801)	0.1949 (0.006*)	0.0779 (0.095)	
Notes.

The objective variable was the median value of the BIS from 10 patients and the explanatory variables were the values of the central frequencies and the TPs of the IMFs in (A) VMD, (B) the EWT, (C) WMD, or (D) VMD+EWT+WMD. The EEG data were obtained from the last 30 min before emergence in 10 patients who received sevoflurane GA.

MAE mean absolute error

RMSE root mean squared error

* p < 0.05.

Figure 5 Multiple linear regression analysis.

Multiple linear regression analysis was performed with the BIS values from the BIS monitor as the objective variable and the statistically significant center frequency (freq) and total power (TP) medians of IMF-1 to IMF-6 obtained by the (A) VMD, (B) EWT, (C) WMD, or (D) VMD+EWT+WMD methods as explanatory variables for the 30 min before emergence in the ten patients (Yamada et al., 2023). (1) The correlation between the observed BIS and predicted BIS is calculated from the multiple regression equation. (2) Visualization of regression coefficients. freq, central frequency, ∗p < 0.05.

Discussion

Assessing the depth of anesthesia through EEG monitoring during GA continues to present numerous challenges, particularly in terms of adapting to the various anesthetic agents and accommodating different patient populations, e.g., children and elderly individuals. The depth of anesthesia indicators that are used in commercially available EEG processing monitors, although useful, are optimized primarily for use with intravenous anesthetics like propofol and inhalational agents such as sevoflurane, which limits their applicability to other general anesthetics. Furthermore, the suitability of these indicators for use with nonadult patients has still to be determined. A better solution is required, and grounding this technology in scientifically validated algorithms may provide the answer. EEG recordings acquired from the frontal region during GA exhibit distinct characteristics, including a reduction in β and γ waves, which are typically associated with wakefulness, because of the effects of the anesthetic agents (Ching et al., 2010; Purdon et al., 2015). In contrast, the hypnotic effects of the GA cause an increase in the slow-wave components, including the δ and θ waves. As the anesthesia intensifies, an increase is also observed in the α waves, which are often linked to sleep spindles. Quantifying these EEG changes will be essential to enable mathematical modeling of the relationship between anesthetic effects and the depth of anesthesia through the frequency dynamics in the EEG signal.

When compared with the Fourier transform, which calculates frequency components and generates spectrograms, the combined use of mode decomposition and the Hilbert transform provides a more precise feature extraction method. This approach captures the IF and IA at each data point, thus enabling generation of high-resolution spectrograms that provide a more representation of the dynamic properties of the signal. In contrast, creation of a power spectrum via the Fourier transform, particularly 0.25 Hz frequency increments for a 128 Hz EEG signal, requires at least 512 data points (equivalent to 4 s of EEG data). This method only achieves a temporal resolution of 2 s under 50% overlap conditions. Therefore, decomposing the EEGs into narrow band IMFs is clearly advantageous because it allows the construction of a Hilbert spectrogram with significantly improved resolution. This technique offers enhanced detail and accuracy in the analysis of time-varying signals.

In our recent investigation, we compared the characteristics of the VMD method (Yamada et al., 2023) with those of the previously reported EMD method (Obata et al., 2023). Although VMD proved effective in capturing the frequency changes that occur in EEGs during GA, the assignment of the narrow band frequency ranges to each IMF varied across the epochs, and this resulted in temporal fluctuations within the frequency bands. These variations represent a limitation when we attempt to establish a consistent relationship between the frequency characteristics of specific IMFs and the depth of anesthesia over time. We also explored alternative mode decomposition methods to address this issue, with a focus on wavelet transform-based techniques. Unlike EMD, the EWT method is grounded within a solid mathematical framework, similar to that of VMD (Dragomiretskiy & Zosso, 2014), which provides a reliable theoretical basis for its application. Although several studies have explored the use of the EWT in the analysis of biological signals such as electrocardiograms and EEGs (Kedadouche, Thomas & Tahan, 2016; Singh & Sunkaria, 2017; Zhuang et al., 2021), its application to EEG analysis during GA has yet to be explored fully.

The Meyer wavelet, which is used in the EWT method, is mathematically well-defined and is shaped specifically to yield precise frequency responses. The effectiveness of the Meyer wavelet as a finite impulse response (FIR) filter is essential because of its ability to analyze signals via convolution operations to isolate the desired frequency components. The impulse response of FIR filters, as suggested by their name, is finite, which means that they respond to an input signal within a finite time. This characteristic is instrumental in the Meyer wavelet’s ability to perform wavelet transformations, a process in which specific frequency components can be extracted from a signal because of the FIR nature of the wavelet filter. The Meyer wavelet’s design ensures that it has a distinct bandwidth within the frequency domain, which is essential for both frequency localization and time domain signal localization. As an FIR filter, the Meyer wavelet adeptly manipulates these components accurately, thus enabling precise extraction of the targeted information from the signals. In the EWT framework, the FIR functionality of the Meyer wavelet is integrated with the mode decomposition, thus enhancing the method’s ability to discern and extract nuanced features from complex signals. This integration not only improves the accuracy of frequency response but also ensures effective signal processing across a variety of the application. To enhance this approach, rather than relying solely on empirical decisions for determination of the boundary frequencies, we implemented a modified version of the EWT method that used a predefined set of boundary frequencies.

Among the frequency-related parameters of the IMFs obtained from mode decomposition methods such as the EWT, WMD, and VMD, those that correlate most strongly with the BIS value—which is the most widely used monitor for depth of anesthesia—may be valuable candidates for assessment of depth of anesthesia and have the potential to match the accuracy of the BIS itself. Accurately capture of EEG frequency changes through computer-based pattern recognition will be crucial for improved anesthesia depth assessments with reliable numerical indicators. In this study, we analyzed EEG frequency characteristics over a 30 min period, from the maintenance phase of GA to awakening by applying three different mode decomposition methods. Both VMD and the EWT showed that the central frequencies of the IMFs were assigned automatically to high-frequency bands that fluctuated across the epochs, irrespective of the transition from anesthesia maintenance to emergence. In the EWT case, the TP of the IMFs in the high-frequency band also varied significantly between epochs. The VMD and EWT methods are beneficial for identification of the primary EEG mode, but they also have limitations. VMD struggles to achieve consistency in the IMF frequency assignment, with variations between epochs and cases, whereas the EWT faces challenges with high-frequency IMFs because of its broad frequency coverage. The WMD method addresses these issues by assigning the IMFs to predefined frequency bands, thus reducing fluctuations, and stabilizing the frequency characteristics. Therefore, WMD offers the potential to provide more stable and predictable results, primarily in cases where the EEG dynamics and the target frequency bands are well understood.

In addition, by imposing constraints on the separation frequency bands, WMD stabilized both the central frequencies and the TP of each IMF across the epochs. This stability provided a smoother, continuous change that correlated well with the BIS values during the 10 min from anesthesia maintenance to awakening. Consequently, using only the VMD or the EWT to derive anesthesia depth indicators has proved challenging because of their high variability of these methods across the epochs. However, WMD yielded IMF patterns that could serve as reliable indicators of the depth of anesthesia. MLR analyses performed with the BIS values showed that the changes observed in the central frequencies of the Hilbert spectra for IMF-1, IMF-3, IMF-5, and IMF-6, along with the TP values for IMF-2, IMF-4, and IMF-6, were significant parameters for use in anesthesia depth evaluation. This customized approach illustrates the potential of WMD to perform more refined and clinically relevant EEG data analyses during anesthesia.

The limitations of this study are tied intrinsically to one fundamental question: “What exactly is the depth of anesthesia?” At present, the BIS and PSi values observed from commercial monitors are used as proxies to measure the depth of anesthesia. However, because the algorithms behind these values are not disclosed, it is difficult to assess their validity by making direct comparisons. This study also investigated the correlation between the parameters from the Hilbert spectra of the IMFs (as obtained via mode decomposition) and the BIS values. The results show that there are inherent challenges involved in identifying anesthesia depth indicators that can outperform BIS, thus highlighting the complexity of this task.

Conclusions

This study investigated the changes in EEG frequency characteristics that occur during the 30 min transition from GA maintenance to awakening via three mode decomposition methods: VMD, the EWT, and WMD. Both VMD and the EWT show variability in the central frequency and TP characteristics, thus making them less reliable options for assessment of the depth of anesthesia. In contrast, WMD maintained stable frequency bands and TP across epochs, with results that correlated well with the BIS values during the 10 min period from anesthesia maintenance to awakening. The WMD-decomposed IMFs showed patterns that could serve as indicators of the depth of anesthesia, and multiple regression analyses revealed significant changes in the central frequencies and the TP characteristics of specific IMFs related to the depth of anesthesia evaluation. WMD outperformed the other methods in terms of capturing the EEG frequency changes that occurred during the transition from anesthesia to wakefulness. The study has highlighted the importance of analyzing any EEG changes during both loss and recovery of consciousness to establish reliable numerical indicators of the depth of anesthesia, particularly in the intermediate states between full wakefulness and unconsciousness. The results obtained showed that the WMD-derived IMFs were consistent across the different patients and tracked BIS index changes effectively during emergence from anesthesia. This indicates that WMD could improve anesthesia monitoring accuracy by refining depth assessment based on subtle EEG frequency characteristics and thus offer a more reliable monitoring tool for anesthesiologists.

Supplemental Information

Supplemental Information 1 Step-by-step explanation of the EWT algorithm

(A) An initial EEG wave (128 Hz, 0.5 s, 64 data points). (B) mirroring the double expansion of the original EEG wave (128 data points). (C) Fast Fourier transform. 1. Power spectrum, 2. Convolutional integral, and 3. Sorted index by argosoft() function. (D) Creation of Mayer filter bank (mfb-0, mfb-1, and mfb-2). (E) Decomposition in the frequency domain. (F) Inverse Fourier transform. 1. Inverse Fourier transform from the frequency domain to the time domain. 2. Un-mirroring to intrinsic mode functions (IMFs), (G). Decomposition into IMFs in time-domains.

Supplemental Information 2 Construction of the Meyer wavelet filter bank in the EWT

(A) ω = ωii=1,2,...,N (N denotes the number of maxima, and also, the number of filter banks.Assuming the frequency domain [0, π] is divided into N consecutive segments, we need to extract N-1 boundaries excluding 0 and π (This figure shows the case of N = 6). (B) To find the boundary in the EWT, local maxima in the spectrum are found and sorted in descending order, and the boundary is defined as the average between consecutive maxima. Let ω n be the limit between each segment (where ω0 = 0 and ωn = π), and denote each segment by Λn = [ωn−1, ωn], then ∪n−1NΛn=0,π. (C) A scheme explaining Eqs. (2) and (3). (D) A tight frame constructed by Meyer’ s wavelet with a set of ϕ1(t), ψn(t)Nn=1 explaining (7). φ(): scale function, ψ (): wavelet function, β: beta function β (x) = x4(35 − 84x + 70x2 + 20x3) ((5)), τ : transition phase, ω : the limit between each segment (where ω0 = 0 and ωn = π). BPF, band-pass filter, IMF, intrinsic mode function; LPF, low-pass filter; Mfb, Meyer wavelet filter bank.

Supplemental Information 3 Hilbert spectrograms of the IMFs and the color density spectral arrays (DSAs) in the VMD, EWT, and WMD methods

IMFs 1–6, a summed signal composed of all IMFs (IMF-all, which is the same as the initial EEG), and color DSAs for 30 min before emergence in all ten patients (as screen capture images from the EEG Mode Decompositor software) are shown.

Supplemental Information 4 Multiple linear regression (MLR) analysis between the BIS values and parameters of IMFs

(A) In the in VMD, EWT, or WMD, using 6 median values of the central frequencies and 6 total powers (TPs) as explanatory variables. (B) Using all parameters of the IMFs derived from the VMD+EWT+WMD as explanatory variables. (C) In the three different mode decomposition using only 6 median values of the central frequencies as explanatory variables, and (D) in the three different mode decomposition using only 6 median values of total powers as explanatory variables. The EEG data were obtained from the last 30 min before emergence in ten patients who received sevoflurane general anesthesia. MAE, mean absolute error; RMSE, root mean squared error; freq, central frequency; TP, total power; ∗p < 0.05.

Supplemental Information 5 Tab-separated values of raw EEG data (microvolts) for 30 min before the emergence of GA in ten patients

Supplemental Information 6 Comma-separated values of processed EEG data

BIS index, SEF95, TP, EMGlow, center frequencies of IMFs, and TPs of IMFs for 30 min before the emergence of GA in ten patients.

Supplemental Information 7 The Jupyter Notebook file of Python (ver. 3.8) code for the EWT

Supplemental Information 8 The implementation Processing (ver. 4.0.5) Ewt and Wmd Class codes in Python

Supplemental Information 9 Video, Patient #6, EMD, 30 min before emergence of GA. A video file showing EMD analysis of EEG data for 30 min leading up to emergence in Patient #6

Supplemental Information 10 Video, Patient #6, VMD, 30 min before emergence of GA. A video file showing VMD analysis of EEG data for 30 min leading up to emergence in Patient #6

Supplemental Information 11 Video, Patient #6, EWT, 30 min before emergence of GA. A video file showing EWT analysis of EEG data for 30 min leading up to emergence in Patient #6

Supplemental Information 12 Video, Patient #6, WMD, 30 min before emergence of GA. A video file showing WMD analysis of EEG data for 30 min leading up to emergence in Patient #6

Supplemental Information 13 MLR analysis of the BIS values and the parameters of the IMFs in VMD, the EWT, or WMD

The objective variable was the median value of the BIS from 10 patients and the explanatory variables were the statistically significant median values of the central frequencies and the TPs of the IMFs in A) VMD, B) the EWT, C) WMD, or D) VMD+EWT+WMD. The EEG data were obtained from the last 30 min before emergence in 10 patients who received sevoflurane GA. MAE: mean absolute error; RMSE, root mean squared error; ∗p < 0.05.

Supplemental Information 14 MLR analysis using the BIS values and the parameters of the IMFs in VMD, the EWT, or WMD

The objective variable was the median value of the BIS from 10 patients, and the explanatory variables were (1) the six median values of the central frequencies and (2) the six total powers of the IMFs. The EEG data were obtained from the last 30 min period before emergence in 10 patients who received sevoflurane GA. MAE, mean absolute error; RMSE, root mean squared error; ∗p < 0.05.

We thank David MacDonald, MSc, from Edanz for editing the draft of this manuscript.

Additional Information and Declarations

Competing Interests

Author Contributions

Human Ethics

Data Availability

The authors declare there are no competing interests.

Shoko Yamochi performed the experiments, analyzed the data, authored or reviewed drafts of the article, and approved the final draft.

Tomomi Yamada performed the experiments, analyzed the data, prepared figures and/or tables, authored or reviewed drafts of the article, and approved the final draft.

Yurie Obata performed the experiments, analyzed the data, prepared figures and/or tables, authored or reviewed drafts of the article, and approved the final draft.

Kazuki Sudo performed the experiments, analyzed the data, authored or reviewed drafts of the article, and approved the final draft.

Mao Kinoshita performed the experiments, analyzed the data, authored or reviewed drafts of the article, and approved the final draft.

Koichi Akiyama performed the experiments, analyzed the data, authored or reviewed drafts of the article, and approved the final draft.

Teiji Sawa conceived and designed the experiments, performed the experiments, analyzed the data, prepared figures and/or tables, authored or reviewed drafts of the article, and approved the final draft.

The following information was supplied relating to ethical approvals (i.e., approving body and any reference numbers):

The Institutional Review Board (IRB) of the Kyoto Prefectural University of Medicine (KPUM) (No. ERB-C-1074).

The following information was supplied regarding data availability:

The raw data and code are available in the Supplemental Files.

The EEG dataset is available at GitHub and Zenodo:

– https://github.com/teijisw/EEG_DataSet

– teijisw. (2024). teijisw/EEG_DataSet: Supplementary Dataset of Wavelet Transform-based Mode Decomposition (v.1.0.0). Zenodo. https://doi.org/10.5281/zenodo.13989502.

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
