# Peer review of "Wavelet transform-based mode decomposition for EEG signals under general anesthesia"

_PeerJ, doi:10.7717/peerj.18518_

## Round 0.1 · original submission · Major Revisions

We have completed the evaluation of your manuscript. Based on the comments and recommendations from the two expert reviewers, I am willing to reconsider your manuscript following major revision and modification. It will be necessary for you to carefully address all the points raised by the 2 reviewers. In particular, the questions by reviewer 2 about the experimental design, validity of the findings, contribution of the findings, and the limitations of the study must be thoroughly and satisfactorily answered before acceptance can be considered.

Reviewer 1 ·

Basic reporting

This paper introduces a study comparing EEG mode decomposition methods (EWT, WMD, and previously reported VMD) to analyze EEG data during sevoflurane anesthesia. The study used EEG Analyzer software to record raw EEG signals from a BIS monitor, followed by mode decomposition using VMD, EWT.
The manuscript uses clear, professional English throughout. However, some sentences could be improved for better readability. For example, in the introduction, "In this study, as a method different from VMD," could be rephrased as "In this study, we employed two alternative methods to VMD."
The figures are relevant, high quality, well-labelled, and described.

Experimental design

The research falls within the journal's scope and addresses a well-defined, relevant, and meaningful question. The study aims to compare different mode decomposition methods to analyze EEG signals during general anesthesia. Ethical standards are maintained, with appropriate IRB approval and patient consent processes described.

Validity of the findings

The underlying data is robust, statistically sound, and well-controlled. The multiple linear regression models and comparisons provide a thorough analysis. The conclusions are well stated, linking back to the original research question and supported by the results. However, the conclusions could be expanded to discuss the clinical implications more thoroughly.

Additional comments

The second part of line 59, starting with 'In this study, as a method different from VMD,' can be introduced as a new subphrase.
Please add an outline of the paper at the end of the Introduction section.
There is a problem with the link: https://anesth- 166 kpum.org/blog_ts/?p=4114
It would be recommended to expand the conclusions of the paper.
A few words mistakenly written: "Figure 5 - Page 28., "beforeemergence”.
Consider adding more details about the limitations of the study and potential directions for future research.

Annotated reviews are not available for download in order to protect the identity of reviewers who chose to remain anonymous.

Reviewer 2 ·

Basic reporting

Over all paper is good. The authors have used the professional English used throughout.
Literature references, sufficient field background provided.
Professional article structure, figures, tables.

Experimental design

What is the primary objective of this study regarding EEG analysis during anesthesia?

How do the authors justify the need for exploring wavelet transform-based mode decomposition methods over previously used methods like VMD?

What are the specific characteristics of the intrinsic mode functions (IMFs) that were analyzed, and why are they important in the context of anesthesia monitoring?

What is the significance of comparing EWT and WMD with VMD in terms of their application to EEG data during anesthesia?

Validity of the findings

What were the main findings regarding the frequency characteristics of IMFs extracted by EWT, WMD, and VMD?
How did the frequency ranges and variances of IMFs differ across the different modes of decomposition?
What did the Hilbert spectrograms reveal about the frequency characteristics of IMFs from EWT and WMD?
Which mode decomposition method demonstrated the strongest correlation with the BIS index, and what were the key metrics for this assessment?

Additional comments

How do the findings of this study contribute to the current understanding of EEG-based anesthesia monitoring?
What limitations were identified in the study, and how might these limitations affect the interpretation of the results?

---

## Round 0.2 · Minor Revisions

Please incorporate the minor changes as suggested by Reviewer 1 in their 'Additional Comments' section.

Reviewer 1 ·

Basic reporting

The article demonstrates a clear improvement in terms of language use, with well-adapted native-level English throughout. The clarity of the text has significantly enhanced the understanding of both the context and the main objective of the study.

Experimental design

The experimental design is well-structured and adheres to high technical and ethical standards. The methods are described now in sufficient detail, providing all necessary information.

Validity of the findings

No comment.

Additional comments

At line 22 in the Abstract, you mentioned: “electroencephalogram (EEG)”, then, at line 46, on the same page (page 5) in the Introduction, you also mentioned: “electroencephalogram (EEG)”. Please revise the content to clarify which one is abbreviated as EEG.
On page 7, it might be a good idea to ensure that the equations are consistently sized. For example, Equation 3 extends to the right compared to Equations 2 and 4. The same issue occurs with Equation 7.
Equations 15 and 16 are not clear; there seems to be a formatting issue. Please revise these two equations.
At Line 190, where you mention delta (0 < f ≤ 4 Hz), please replace the value 0 with 0.5. Statistically, 0.5 is the correct value when referencing EEG recordings.
In Figure 1, there are some writing errors:
1. In Step 4, separate the words: “thefrequency” into “the frequency”.
2. In Step 5, replace “boudary” with “boundary”.
Additionally, please review the content of Figure 1 for other potential typographical mistakes.

Annotated reviews are not available for download in order to protect the identity of reviewers who chose to remain anonymous.

Reviewer 2 ·

Basic reporting

The authors have addressed all the comments; however, similar work already exists in the literature.

Experimental design

Properly provided all the details in the revised manuscript.

Validity of the findings

Properly provided all the details in the revised manuscript.

Additional comments

There are no further comments.

---

## Round 0.3 · accepted · Accept

I am satisfied that all the reviewers' comments have been satisfactorily addressed, and can confirm that the article is now suitable for publication.